# Discovery of Major Quantitative Trait Loci and Candidate Genes for Fresh Seed Dormancy in Groundnut

Deekshitha Bomireddy [1,2], Sunil S. Gangurde [1,3,4], Murali T. Variath [1], Pasupuleti Janila [1], Surendra S. Manohar [1], Vinay Sharma [1], Sejal Parmar [1], Dnyaneshwar Deshmukh [1], Mangala Reddisekhar [2], Devarapalli Mohan Reddy [5], Palagiri Sudhakar [2], Bommu Veera Bhaskara Reddy [5], Rajeev K. Varshney [1,6], Baozhu Guo [4] and Manish K. Pandey [1,*]

1　International Crops Research Institute for the Semi-Arid Tropics (ICRISAT), Hyderabad 502324, India; b.deekshitha@cgiar.org (D.B.); Sunil.Gangurde@uga.edu (S.S.G.); murali.tv2006@gmail.com (M.T.V.); p.janila@cgiar.org (P.J.); m.surendra@cgiar.org (S.S.M.); s.vinay@cgiar.org (V.S.); p.sejal@cgiar.org (S.P.); d.deshmukh@cgiar.org (D.D.); rajeev.varshney@murdoch.edu.au (R.K.V.)
2　S.V. Agricultural College, ANGRAU, Tirupati 517502, India; mreddisekhar11@gmail.com (M.R.); sudhakarpalagiri@gmail.com (P.S.)
3　Crop Protection and Management Research Unit, USDA-ARS, Tifton, GA 31793, USA
4　Department of Plant Pathology, University of Georgia, Tifton, GA 31793, USA; baozhu.guo@usda.gov
5　Institute of Frontier Technology, RARS, ANGRAU, Tirupati 517502, India; dmrgene@gmail.com (D.M.R.); bvbreddy68@gmail.com (B.V.B.R.)
6　State Agricultural Biotechnology Centre, Centre for Crop & Food Innovation, Food Futures Institute, Murdoch University, Murdoch 6150, Australia
*　Correspondence: m.pandey@cgiar.org; Tel.: +91-40-3071-3305

**Abstract:** Spanish bunch groundnut varieties occupy most of the cultivated area in Asia and Africa, and these varieties lack required 2-3 weeks of fresh seed dormancy (FSD) hampering kernel quality. Genomic breeding can help to improve commercial groundnut cultivars for FSD in a shorter time with greater precision. In this regard, a recombinant inbred line (RIL) population from the cross ICGV 02266 (non-dormant) × ICGV 97045 (dormant) was developed and genotyped with a 5 K mid-density genotyping assay. A linkage map was constructed with 325 SNP loci spanning a total map length of 2335.3 cM and five major QTLs were identified on chromosomes Ah01, Ah11, Ah06, Ah16 and Ah17. Based on differential gene expression using transcriptomic information from dormant (Tifrunner) and non-dormant (ICGV 91114) genotypes, *histone deacetylases*, *histone-lysine N-methyltransferase*, *cytochrome P450*, *protein kinases*, and *ethylene-responsive transcription factor* were identified as key regulators involved in the hormonal regulation of dormancy. Six Kompetitive Allele Specific PCR (KASP) markers were successfully validated in the diverse panel including selected RILs of the same population and germplasm lines. These validated KASP markers could facilitate faster breeding of new varieties with desired dormancy using marker-assisted early generation selection.

**Keywords:** genetic map; KASP markers; mid-density genotyping assay; pre-harvest sprouting

## 1. Introduction

　　Groundnut or peanut (*Arachis hypogaea* L.) is a widely grown oilseed crop, that belongs to the family of Leguminosae or Fabaceae. It is known for its diversified uses, such as cooking oil, food, confectionary, and dietary uses for human consumption, as well as fodder for livestock [1]. Global production of groundnut is 66.3 million tons, covering 34.1 million hectares of cultivated area making it the third-largest important oilseed crop [2]. There are certain widespread constraints affecting groundnut yield and quality. In situ germination of freshly matured seeds at the time of harvesting is one such potential constraint limiting the production as well as end use quality of groundnut kernels. 60% of total groundnut production occurs in the semi-arid regions of Africa and Asia, where Spanish varieties are typically grown by marginal farmers under rainfed conditions [3].

Generally, Spanish genotypes are early maturing and lack fresh seed dormancy (FSD), unlike Virginia genotypes, which have a long maturity duration with variable periods of dormancy [4,5]. Therefore, upon erratic rainfall prior to harvesting the crop, seeds of Spanish genotypes tend to germinate in pods inside the soil, leading to reduced yield and hampered quality. In addition, these pre-germinated seeds are prone to pathogen infection and contamination, which reduces their market price, resulting in economic losses to the farmers. Moreover, a seed is a connecting link between two consecutive generations regulated by two important events called dormancy and germination. A longer dormancy or delayed germination increases the length of the crop season and reduces the number of generations per year [6]. Therefore, the presence of 2–3 weeks of FSD in the early-maturing popular groundnut cultivars is desirable in order to overcome this problem. Though foliar application of growth inhibitors like maleic hydrazide has been known to induce dormancy in Spanish types [7], it is not a cost-effective approach, especially for rainfed cultivation. So, breeding for varieties with shorter periods of FSD is a sustainable and economically feasible option. Such varieties are a boon to farmers as these allow delayed harvesting of the crop at times of unseasonal rains without any preharvest sprouting losses.

In conventional breeding, breeding lines and segregating populations for FSD are selected and forwarded on the basis of germination tests carried out under controlled conditions in a seed incubator. On the other hand, genomics-assisted breeding (GAB) has proven its potential by improving various traits in groundnut and other leguminous crops [8,9]. GAB benefits from the optimal utilization of resources and time with the development of linked diagnostic markers. Identification of genomic regions and candidate genes associated with FSD can help in the development of user-friendly markers for efficient introgression of FSD QTLs/genes into important groundnut cultivars. So far, very few attempts have been made to map the QTLs or genes associated with FSD in groundnut. One such attempt was made in a $F_2$ population of ICGV 00350 (non-dormant) × ICGV 97045 (dormant), which was genotyped using Diversity Arrays Technology (DArT) and DArT-seq markers and reported two QTLs, *qfsd−1* and *qfsd−2* with 22.14% and 71.21% of PVE, respectively [3]. As of late, with the availability of draft genome sequences [10,11] and advancement in sequencing-based approaches, the QTL-seq approach was utilized in the RIL population developed from the previous cross [3] and identified two genomic regions on A09 and B05 and *zeaxanthin epoxidase* and *RING-H2 finger protein* as the candidate genes responsible for inducing FSD, and developed the GMFSD1 marker, which was validated in several germplasm lines [12]. Development of groundnut 58 K SNP array [13] accelerated trait mapping studies in groundnut and was successfully used to map several yield related traits [14,15]. More recently, by genotyping a RIL population using the 'Axiom_*Arachis*' 58 K SNP array, two QTLs for FSD were identified on the A04 and A05 chromosomes with 43.16% and 51.61% of PVE, respectively [16]. Advancements in sequencing technologies have drastically reduced the cost of sequencing, and three tetraploid genome assemblies for three different genotypes of cultivated groundnut have been developed [17–19]. In the present study, we used newly developed highly informative and cost-effective mid-density SNP assay of 5 K SNPs (ICRISAT, Unpublished).

Though germinability or dormancy are genetically controlled traits, they are also influenced by environmental factors such as moisture, humidity, light, and temperature during crop maturity [20]. Plant growth regulators such as abscisic acid (ABA) have been known to positively regulate dormancy, whereas gibberellins (GA) and ethylene tend to break the dormancy [21]. Environmental factors tend to modulate ABA/GA balance by altering their metabolic and signaling pathways [22,23]. In different segregating populations ($F_2$), the inheritance ratio of dormant to non-dormant lines were reported to be 3:1, making the dormancy allele dominant with monogenic or qualitative inheritance [5,24]. On the contrary, results from other $F_2$ populations showed that the trait is segregating in a 15:1 ratio (non-dormant:dormant) indicating that the trait is quantitatively regulated by two duplicate recessive genes [3,4]. From inconsistent and conflicting conclusions from previous studies, seed dormancy in groundnut appears to be a complicated trait as the

number and nature (recessive or dominant) of genes governing the trait depend on the parents or cultivars used for the development of respective populations [16]. Therefore, for a clear understanding of the trait and to develop reliable markers, extensive research has to be done in diverse genetic backgrounds to identify additional QTLs and candidate genes. In this context, a RIL population ICGV 02266 (non-dormant) × ICGV 97045 (dormant) has been developed and genotyped using a 5 K SNP mid-density assay. We have identified robust QTLs with high phenotypic variance for FSD. Our study can facilitate efficient introgression of the trait, with the help of diagnostic markers, to improve FSD in important groundnut cultivars.

## 2. Materials and Methods

### 2.1. Development of RIL Population ICGV 02266 × ICGV 97045 and Phenotyping for Fresh Seed Dormancy

A RIL population comprising of 160 $F_{7:8}$ lines was developed by using the single seed descent method from the cross ICGV 02266 (non-dormant) × ICGV 97045 (dormant) at ICRISAT, Patancheru, Hyderabad (Figure 1). ICGV 02266 is a non-dormant, short-duration Spanish bunch (subspecies *fastigiata*; botanical type *vulgaris*) derived from ICGV 94143 × ICGV 94136 parentage. It is also a drought tolerant, bold seeded, dual-purpose variety (higher pod and fodder yield) suitable for the Rabi-Summer plantation. Under good cultural conditions, it has recorded a yield of 4167 kg/ha with 72% shelling out turn and 50% oil content, but it lacks FSD. Whereas, ICGV 97045 is a dormant, Virginia variety and has up to 15 days of dormancy, so it is used as donor parent to develop the RIL population.

Phenotyping for FSD was carried out as per the methodology given by [5] for all the 160 RILs along with their parents during post-rainy 2018–2019 (S1), post-rainy 2019–2020 (S2) and rainy 2021 (S3). Immediately after harvesting, from each RIL, 10 good quality matured seeds were selected and treated with fungicide (Mancozeb and Carbendazim) and placed on moist germination paper in a petri dish for an in vitro germination assay with two replicates. The petri dishes were kept in the incubator at 37 °C in complete darkness. To maintain moisture on the germination paper, regular water was sprayed at 24 hr intervals. Seed germination count of each genotype was recorded daily for 15 days to calculate germination rate. The time required to achieve 50% germination for a line is referred to as the dormancy period of that particular line. Lines that showed 50% germination were considered non-dormant lines, and lines that took more than 7 days to achieve 50% germination were referred to as dormant lines.

### 2.2. DNA Extraction and Genotyping with 5 K SNP Assay

Tender leaf samples from all the 160 RILs and two parental genotypes were collected, after 30 days of sowing, for DNA isolation. The DNA was extracted using the Nucleospin Plant II kit (Macherey–Nagel, Duren, Germany). The DNA quantity and quality were checked on a 0.8% agarose gel and Nanodrop 8000 Spectrophotometer (Thermo Fisher Scientific Inc., Waltham, MA, USA), respectively. A total of 20 µL of DNA with a concentration of 20 ng/µL from each RIL along with parents were used for genotyping at Thermo Fisher's Agri-Seq Targeted GBS platform, USA with a mid-density assay of 5 K SNPs.

### 2.3. Genetic Map Construction

The 5 K SNPs were first checked for parental polymorphism, followed by a chi-square ($\chi 2$) test to check the segregation ratio (1:1) of SNPs for genetic map construction. Allelic calls for each SNP in the population were recorded based on the parents, the recurrent (non-dormant) ICGV 02266 type calls (AA) were represented as "2", and donor (dormant) ICGV 97045 type calls (BB) were represented as "0", heterozygotes (AB) were represented by "1" and "−1" was assigned to missing (-). After filtering out monomorphic, unlinked, and highly distorted SNPs, the remaining SNPs were then used for genetic linkage map construction using QTL IciMapping version 4.2 software [25]. To convert recombination frequencies (RF) into map distances centiMorgans (cM), Kosambi map

function was used [26]. Grouping of markers into linkage groups was done by applying a recombination frequency (∂) threshold of 30% followed by ordering and rippling. The best linkage map with a suitable map length according to genome size was constructed by adding, rearranging or deleting the marker functions implemented in ICIM.

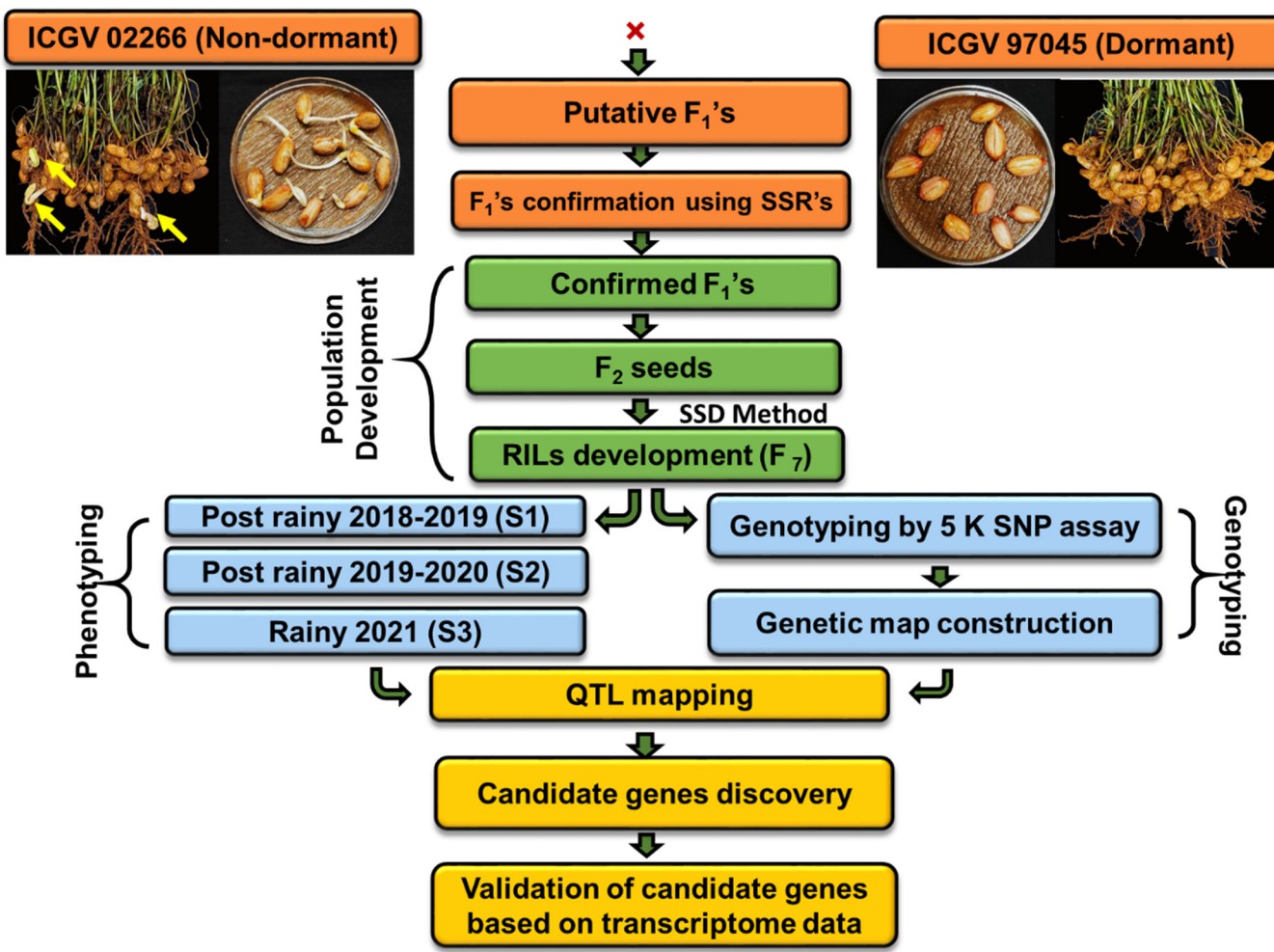

**Figure 1.** Flow chart of RIL population development, genotyping, and phenotyping for fresh seed dormancy. The RIL population (ICGV 02266 × ICGV 97045) was developed using the single seed decent (SSD) method followed by three-season phenotyping, genotyping, QTL mapping, candidate gene discovery, and differential gene expression analysis using transcriptome data of dormant Tifrunner and non-dormant ICGV 91114.

*2.4. Quantitative Trait Loci Analysis*

Phenotypic data from three seasons (S1, S2 and S3) along with genotyping data and genetic map information were used to identify the QTLs associated with FSD. Initially, Inclusive Composite Interval Mapping (ICIM) for QTLs with additive (ICIM-ADD) effect was performed in QTL IciMapping version 4.2 software [25] with a 1 cM scanning step and 0.001 as the $p$ value for entering variables (PIN) in a stepwise regression. The threshold LOD was determined by 1000 permutations at a $p < 0.05$ significance level. A LOD threshold score of 3 was set to declare the presence of significant QTLs associated with the trait. To confirm the results obtained from ICIM mapping, we also conducted composite-interval mapping (CIM) using WinQTL Cartographer version 2.5 [27], where window size, walking speed, and number of control markers were set to 10 cM, 1 cM, and 5 cM, respectively. The LOD threshold to determine the significant QTL was established with 1000 permutations at $p \leq 0.05$ [28]. In both cases, QTLs with a LOD score of $\geq 3$ and >10% of PVE explained were

considered to be major QTLs associated with FSD. For better visualization, the linkage map and QTLs were drawn using MapChart version 2.3 [29].

### 2.5. Epistatic (Q × Q) QTL Analysis

We performed epistatic QTL analysis to identify the combined effect of two QTLs on the FSD trait. Phenotyping data along with genetic map information was used to identify the epistatic QTLs for FSD using QTL IciMapping version 4.2 software [25]. For epistatic analysis, ICIM for digenic epistatic QTLs (or two-dimensional ICIM, ICIM-EPI) method with 5 cM scanning step and 0.001 *p*-value for entering variables (PIN) in stepwise regression was used. A LOD threshold score of 3 was used to establish significant epistatic QTLs.

### 2.6. Identification of Candidate Genes and Validation of KASP Markers for FSD

In the present study, candidate genes were identified in the major QTL regions identified for FSD. The genomic region between the physical positions of the flanking markers of each QTL was used to investigate the candidate genes on the groundnut genome on Peanut-Base [https://peanutbase.org/ (accessed on 10 October 2021)]. To confirm the role of the identified candidates in the dormancy/germination processes in groundnut, we checked the expression levels of candidate genes in *Arachis hypogaea*, both sub spp. *fastigiata* and *hypogaea* gene expression atlases (AhGEA) available in PeanutBase [https://peanutbase.org/ (accessed on 10 October 2021)]. A gene expression atlas was developed using 22 tissues at various developmental stages for Tifrunner, a dormant cultivar of subspp. *hypogaea* [30]. Similarly, another gene expression atlas was developed using more than 20 tissues at various seed and pod developmental stages for ICGV 91114, a non-dormant and drought tolerant cultivar of subspp. *fastigiata* [31]. From both atlases, we used expression values from six tissues selected from seed and pod developmental stages. Genes that showed contrasting expression in dormant and non-dormant cultivars were considered as the potential candidates that may be playing an important role in the regulation of FSD in groundnut.

From the reported genomic regions of our previous sequencing-based trait mapping approach [12], KASP (Kompetitive Allele Specific PCR) markers were designed. While designing the KASP assays for each SNP marker, two allele-specific forward primers and one common reverse primer were designed at Intertek Pvt. Ltd., Hyderabad, India. The developed KASP markers were validated on a panel comprised of dormant RILs from the population used in this study (ICGV 02266 × ICGV 97045) and germplasm lines from ICRISAT (Table S1).

### 3. Results

### 3.1. Phenotypic Variability for Fresh Seed Dormancy in RIL Population

Three seasons of phenotyping data were recorded for FSD using an in vitro germination assay. In the first 24 h, the germination rates of the parental genotypes, ICGV 02266 (non-dormant) and ICGV 97045 (dormant) were observed to be 50% and 0%, respectively. ICGV 02266 showed a 100% germination rate within 48 hrs. In contrast, ICGV 97045 showed 0% germination up to 15 days in both replications across the seasons. Among 160 RILs, on average 144 RILs showed a 50% germination rate in the first 1–4 days. However, 10 RILs showed a 50% germination rate in 8–10 days, while 6 RILs did not germinate even after 15 days. Normal distribution of the phenotypic data in all three seasons indicated that the trait is quantitative in nature (Figure 2).

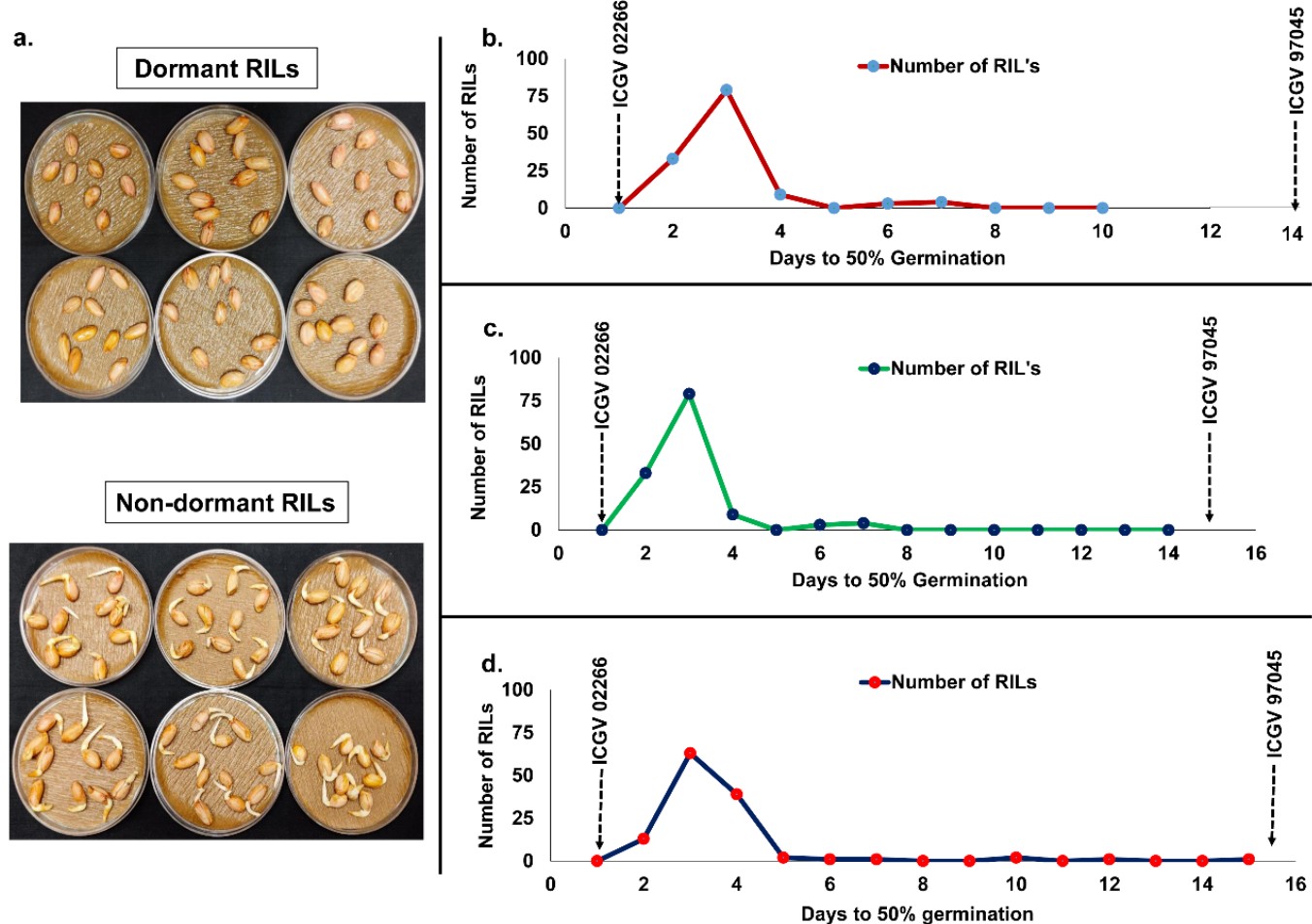

**Figure 2.** Fresh seed dormancy phenotypic variability in the RIL population (ICGV 02266 × ICGV 97045). Pictures of (**a**) dormant RILs 15 days after the FSD in vitro germination assay and (**b**) Non-dormant RILs 24 h after the FSD in vitro germination assay. The line graph represents phenotyping data of the RIL population along with dormant and non-dormant parents during (**b**) Post-rainy 2018–2019 (S1) and (**c**) Post-rainy 2019–2020 (S2) and (**d**) Rainy 2021 (S3).

*3.2. Genetic Map Based on 5 K SNP Mid-Density Assay*

Based on the genotyping data of the parents and 160 RILs generated from the mid-density 5 K assay, out of 5078 SNPs, 4296 SNPs were found to be monomorphic between the parents, and 238 SNPs were missing both parents. So, a total of 544 SNPs were found to be polymorphic between the parents. Of these 544 SNPs, 5 were polymorphic in parents but were monomorphic in the RIL population and 145 SNPs showed high distortion (>85%) based on the chi-square test. Finally, after rigorous filtration, a total of 395 high quality polymorphic SNPs were used for genetic map construction.

Out of 395 SNPs, 70 SNPs were found to be distorted and could not be mapped onto the linkage groups. Finally, a genome-wide linkage map was developed with 325 SNP loci spanning a total map length of 2335.3 cM. The length of individual linkage groups ranged from 45.2 cM (Ah10) to 195.0 cM (Ah03). The number of mapped SNP loci per individual linkage group ranged from 4 (Ah11) to 43 (Ah02). The average inter-marker distance per individual linkage group varied from 1.8 cM (Ah04) to 21.0 cM (Ah07), with a total average inter-marker distance of 7.2 cM per loci (Table 1). The highest number of SNP loci were mapped on the Ah02 linkage group with 43 loci which was the densest linkage group among all 20 chromosomes. However, only four SNP loci were mapped in Ah11 and five in the Ah06 linkage group.

**Table 1.** Summary of the genetic linkage map with 325 mapped SNP loci in RIL population (ICGV 02266 × ICGV 97045).

| Linkage Group | No. of Polymorphic Loci | No. of Mapped Loci | Map Distance (cM) | Average Inter-Marker Distance (cM/loci) |
|---|---|---|---|---|
| Ah01 | 14 | 10 | 143.8 | 14.4 |
| Ah02 | 43 | 43 | 148.1 | 3.4 |
| Ah03 | 29 | 24 | 195.0 | 8.1 |
| Ah04 | 35 | 27 | 49.4 | 1.8 |
| Ah05 | 14 | 11 | 46.9 | 4.3 |
| Ah06 | 10 | 5 | 65.3 | 13.1 |
| Ah07 | 13 | 8 | 167.9 | 21.0 |
| Ah08 | 12 | 10 | 127.0 | 12.7 |
| Ah09 | 12 | 10 | 71.3 | 7.1 |
| Ah10 | 15 | 11 | 45.2 | 4.1 |
| Ah11 | 7 | 4 | 54.1 | 13.5 |
| Ah12 | 31 | 29 | 171.5 | 5.9 |
| Ah13 | 15 | 13 | 125.2 | 9.6 |
| Ah14 | 34 | 24 | 140.3 | 5.8 |
| Ah15 | 19 | 15 | 164.5 | 11.0 |
| Ah16 | 47 | 42 | 188.7 | 4.5 |
| Ah17 | 10 | 9 | 104.1 | 11.6 |
| Ah18 | 8 | 8 | 112.1 | 14.0 |
| Ah19 | 17 | 14 | 166.6 | 11.9 |
| Ah20 | 10 | 8 | 48.2 | 6.0 |
| Total | 395 | 325 | 2335.3 | 7.2 |

*3.3. Quantitative Trait Locus Analysis Identified Five Major Main-Effect QTLs*

QTL analysis resulted in the identification of five major QTLs associated with FSD using the inclusive composite interval mapping additive (ICIM-ADD) and composite interval mapping (CIM) method in ICIM and QTL Cartographer, respectively. These five QTLs were mapped on Ah01, Ah11, Ah06, Ah16, and Ah17 linkage groups with a LOD score ranging from 8.5–19.2 and 10.5–19.6 and phenotypic variance explained (PVE) ranging from 53.2–60.4% and 69.3–74.7% in QTL Cartographer and ICIM, respectively. It is interesting to note that four major main-effect QTLs were mapped on two genomic regions of the homeologous chromosomes, i.e., Ah01/Ah11, and Ah06/Ah16.

In the cases of the QTLs identified by ICIM, QTL '*qfsd_Ah01*' on linkage group Ah01 was identified in the genomic regions flanked by the SNP loci Ah01_44483417—Ah01_39832847 with the highest LOD and PVE of 16.0 and 70.2%, respectively, with an additive effect of −1.9. During S2, QTL '*qfsd_Ah06*' (Ah06_112890279—Ah06_109915576) on Ah06 was identified with 10.5 LOD, 70.1% PVE, and an additive effect of 1.9. Similarly, '*qfsd_Ah16*' (Ah16_141375358—Ah16_138472734) QTL on Ah16 linkage group was also detected during S2 with a LOD score of 12.7, PVE of 70.1% and an additive effect of 1.9. The QTL '*qfsd_Ah17*' (Ah17_69164615—Ah17_2853004) located on Ah17, identified in all three seasons (S1, S2 and S3) with a LOD of 12.8 to 19.6, PVE% of 69.3 to 74.7% and additive effect of −1.9 to −4.2.

The QTLs detected in ICIM were also detected in Cartographer with minor variations in LOD and PVE values. For instance, the QTL '*qfsd_Ah01*' (Ah01_44483417—Ah01_39832847) was identified with a higher LOD of 13.5, 53.9% PVE and an additive effect of −1.7 in S2. The QTL '*qfsd_Ah16*', flanked by Ah16_141375358—Ah16_138472734 was identified in S2 with a LOD of 10.3 and 53.9% PVE and 1.7 additive effect. A QTL '*qfsd_Ah17*' (Ah17_69164615—Ah17_2853004) was identified both in S1 and S3 with 12.9–19.2 LOD, 53.2–60.4% PVE and an additive effect of −2.0 to −4.0. A QTL '*qfsd_Ah06*' flanked by Ah06_112890279—Ah06_109915576 with LOD score of 8.5 and 53.9% of PVE and an additive effect of 1.7 was identified in S2. A QTL, *qfsd_Ah11* was identified on chromosome Ah11 with 12.9 LOD and 53.9% PVE and an additive effect of −1.7 during PR 2019–2020

(Table 2; Figure 3). In addition, few QTLs were identified on chromosome Ah02, Ah04 and Ah15 with 11.2–63.2 PVE% with LOD of 3.2 to 24.6 (Table S2).

**Table 2.** Major effect QTLs identified for FSD in ICGV 02266 × ICGV 97045 using QTL Cartographer and ICIM.

| QTL Name | Chr | Season | Left Marker | Right Marker | QTL Cartographer | | | | ICIM | | | |
|---|---|---|---|---|---|---|---|---|---|---|---|---|
| | | | | | Pos (cM) | LOD | PVE (%) | Add | Pos (cM) | LOD | PVE (%) | Add |
| *qfsd_Ah01* | Ah01 | S2 | Ah01_44483417 | Ah01_39832847 | 100 | 13.5 | 53.9 | −1.7 | 100 | 16 | 70.2 | −1.9 |
| *qfsd_Ah11* | Ah11 | S2 | Ah11_8690364 | Ah11_17741529 | 49 | 12.9 | 53.9 | −1.7 | - | - | - | - |
| *qfsd_Ah06* | Ah06 | S2 | Ah06_112890279 | Ah06_109915576 | 13 | 8.5 | 53.9 | 1.7 | 13 | 10.5 | 70.1 | 1.9 |
| *qfsd_Ah16* | Ah16 | S2 | Ah16_141375358 | Ah16_138472734 | 181 | 10.3 | 53.9 | 1.7 | 180 | 12.7 | 70.1 | 1.9 |
| *qfsd_Ah17* | Ah17 | S1 | Ah17_69164615 | Ah17_2853004 | 94 | 12.9 | 53.2 | −2.0 | 94 | 16.2 | 69.3 | −2.1 |
| *qfsd_Ah17* | Ah17 | S2 | Ah17_69164615 | Ah17_2853004 | - | - | - | - | 94 | 12.8 | 70.1 | −1.9 |
| *qfsd_Ah17* | Ah17 | S3 | Ah17_69164615 | Ah17_2853004 | 94 | 19.2 | 60.4 | −4.0 | 94 | 19.6 | 74.7 | −4.2 |

**S1:** Post rainy 2018–2019; **S2:** Post rainy 2019–2020; **S3:** Rainy 2021; **LOD:** Logarithm of odds; **PVE:** Phenotypic variance explained; **Chr:** Chromosome; **Pos:** Position; **cM:** CentiMorgan; **Add:** Additive effect.

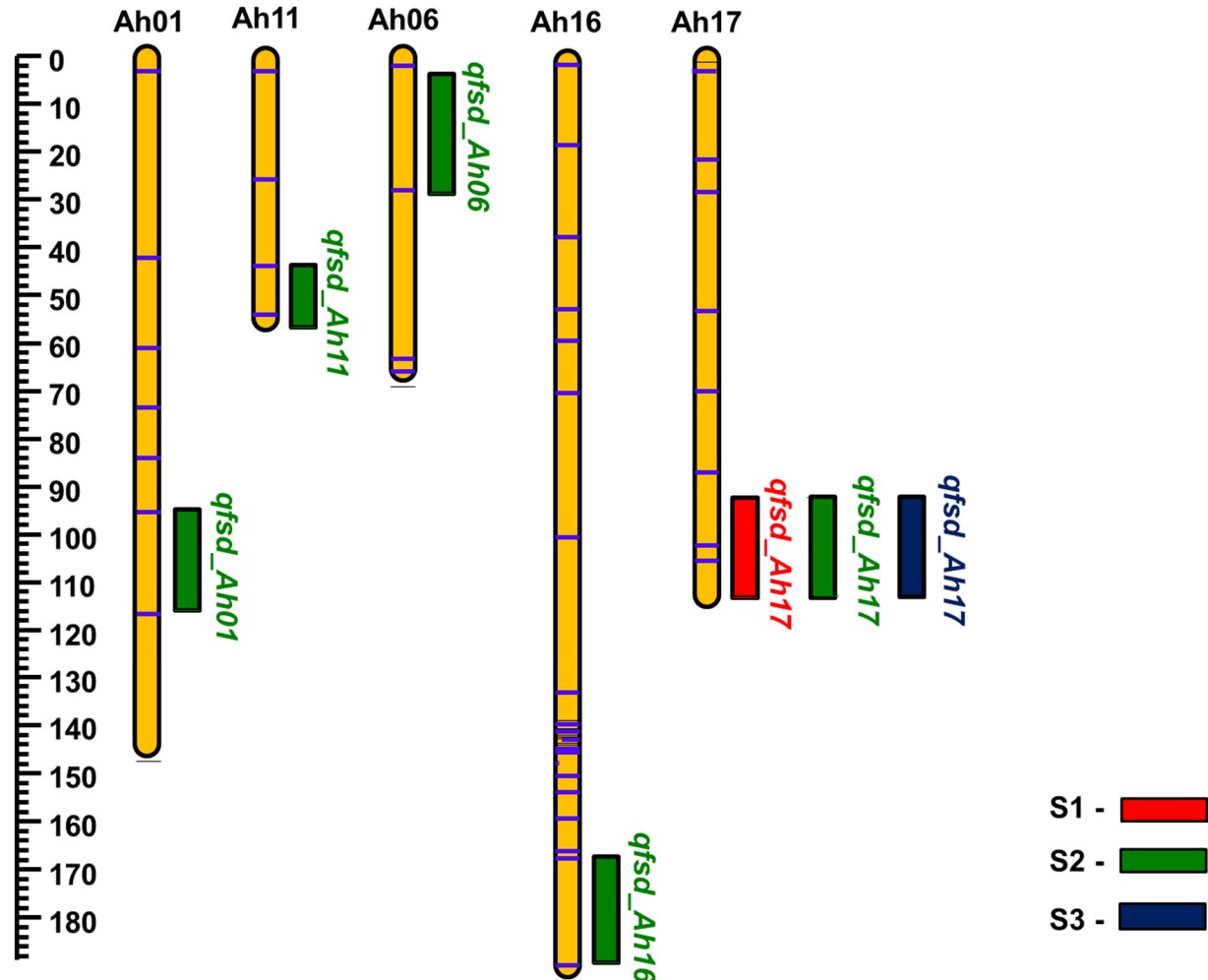

**Figure 3.** Major QTLs associated with fresh seed dormancy.

### 3.4. Epistatic QTLs (QTL × QTL Interaction)

Epistatic analysis detected a total of 103 epistatic QTLs (QTL × QTL interaction) with LOD score and PVE ranging from 4.5–29.0 and 26.5–92.3%, respectively. Out of 103 epistatic-QTLs, 34 were detected in S1, 35 were detected in S2 and 34 were identified in S3. Three epistatic-QTLs were consistently detected in all the three seasons, with PVE and LOD ranging from 68.6–75.5% and 19.0–27.6, respectively (Table 3). We observed that the majority of epistatic interactions were identified on homeologous chromosomes such

as, Ah02/Ah12, Ah03/Ah13, Ah06/Ah16 and Ah10/Ah20 with higher LOD and PVE% (Table S3).

**Table 3.** Epistatic QTLs (Q × Q) consistently identified in S1, S2, and S3 for FSD.

| Season | Chr 1 | Pos 1 | Left Marker 1 | Right Marker 1 | Chr 2 | Pos 2 | Left Marker 2 | Right Marker 2 | LOD | PVE (%) | Add 1 | Add 2 | Add by Add |
|---|---|---|---|---|---|---|---|---|---|---|---|---|---|
| S1 | 5 | 25 | Ah05_8320341 | Ah05_112121360 | 8 | 60 | Ah08_10460625 | Ah08_37701095 | 19.0 | 69.2 | −1.1 | −1.1 | 1.1 |
| S2 | 5 | 20 | Ah05_8320341 | Ah05_112121360 | 8 | 60 | Ah08_10460625 | Ah08_37701095 | 20.9 | 71.0 | −0.9 | −1.0 | 1.0 |
| S3 | 5 | 15 | Ah05_8320341 | Ah05_112121360 | 8 | 60 | Ah08_10460625 | Ah08_37701095 | 29.0 | 74.8 | −2.0 | −2.0 | 2.1 |
| S1 | 12 | 140 | Ah12_1309953 | Ah12_40605934 | 14 | 35 | Ah14_134748938 | Ah14_133651188 | 19.6 | 68.6 | 1.2 | −1.1 | −1.2 |
| S2 | 12 | 140 | Ah12_1309953 | Ah12_40605934 | 14 | 30 | Ah14_134748938 | Ah14_133651188 | 21.3 | 74.2 | 1.0 | −1.0 | −0.8 |
| S3 | 12 | 140 | Ah12_1309953 | Ah12_40605934 | 14 | 40 | Ah14_134748938 | Ah14_133651188 | 27.6 | 75.5 | 2.0 | −2.2 | 2.1 |
| S1 | 12 | 75 | Ah12_20615086 | Ah12_114188707 | 19 | 40 | Ah19_1302657 | Ah19_11030104 | 21.1 | 68.7 | 1.2 | 1.1 | 1.2 |
| S2 | 12 | 80 | Ah12_20615086 | Ah12_114188707 | 19 | 40 | Ah19_1302657 | Ah19_11030104 | 22.3 | 70.5 | 0.9 | 0.9 | 1.0 |
| S3 | 12 | 80 | Ah12_20615086 | Ah12_114188707 | 19 | 40 | Ah19_1302657 | Ah19_11030104 | 22.0 | 75.0 | 2.0 | 2.1 | 2.1 |

**S1:** Post rainy 2018–2019; **S2:** Post rainy 2019–2020; **S3:** Rainy 2021; **Chr:** Chromosome; **Pos (cM):** Position (CentiMorgan); **LOD:** Logarithm of odds; **PVE:** Phenotypic variance explained; **Add:** Additive effect; **Add by Add:** Additive effect by additive effect.

*3.5. Identification of Candidate Genes and Validation of KASP Markers*

For the '*qfsd_Ah01*' QTL, a 4.6 Mb genomic region was identified on Ah01 with flanking markers Ah01_44483417 and Ah01_39832847. Evaluating this 4.6 Mb region with groundnut genome assembly [(https://peanutbase.org/ (accessed on 10 October 2021)] accompanied by genome annotation data resulted in the identification of four potential candidate genes such as *AP2-like ethylene-responsive transcription factor, L-lactate dehydrogenase A-like protein* which are involved in ABA/GA/ethylene signaling regulating either dormancy or germination. Similarly, a 9.0 Mb region on chromosome Ah11 for the QTL '*qfsd_Ah11*' flanked in between Ah11_8690364 and Ah11_17741529 was comprised of 30 potential candidate genes like *histone deacetylase 1, zinc finger CCCH-type with G patch domain protein, RING-H2 finger protein 2B (zinc finger, RING/FYVE/PHD-type)*, which have been reported to regulate seed dormancy/germination in many crops. Genome annotation data from 2.9 Mb genomic regions of each QTL '*qfsd_Ah06*' (Ah06_112890279—Ah06_109915576) and '*qfsd_Ah16*' (Ah16_141375358—Ah16_138472734) resulted in the identification of 22 and 16 potential candidate genes, respectively. *WRKY family transcription factor, protein phosphatase 2C family protein, and histone-lysine N-methyltransferase ATX5-like* were among the most well-known for regulating seed dormancy. For QTL '*qfsd_Ah17*' (Ah17_69164615—Ah17_2853004), a 1 Mb genomic region was considered on Ah17 to identify genes associated with dormancy and successfully identified 25 potential candidate genes. The Ah17 chromosome was found to have an *ethylene-responsive transcription factor*, as well as multiple copies of the *Cytochrome P450 superfamily protein* and a *receptor-like protein kinase 2 (Leucine-rich repeat)*. Altogether, 97 potential candidate genes (Table S4) which are known to be involved in ABA/GA/ethylene biosynthesis or signaling pathways were identified as the candidates regulating dormancy/germination.

To further understand the expression pattern of the above identified genes and to confirm the key genes associated with FSD in groundnut, the expression levels of these genes in ICGV 91114 (non-dormant) were compared with those of Tifrunner (dormant). As a result, 25 genes were found to be expressed contrastingly in at least one of the six selected tissues at key developmental stages (Table 4 and Table S5; Figure 4). As both the cultivars differ in their dormancy status, genes that showed contrasting expression in both the cultivars (ICGV 91114 and Tifrunner) were considered as the potential candidates regulating groundnut FSD. Higher expression levels of *histone deacetylase 1* were observed in selected tissues of non-dormant ICGV 91114 but *histone deacetylase 2* expression was higher in the seeds and embryo of dormant Tifrunner. Receptor-like protein kinases showed higher expression pattern at all stages of seed and pod in Ah11, but they showed lower expression in Ah06 and Ah16 of dormant Tifrunner. *Eukaryotic aspartyl protease* and *MYB transcription factor MYB60* were noticed to be highly expressed in the pod wall of ICGV 91114. Similarly, *Cytochrome P450 superfamily protein genes, histone-lysine N-methyltransferase ATX5-like, protein phosphatase 2C* and *serine/threonine-protein kinase SRK2I-like isoform 1* have higher expression values in almost all the six tissues of ICGV 91114. These systematically

identified genes can be further analyzed and deployed for the development of diagnostic markers for improving the FSD trait.

**Table 4.** Genes associated with FSD identified from major QTL regions showing contrasting expression between dormant Tifrunner and non-dormant ICGV 91114.

| QTL | Chromosome | Left Marker | Right Marker | Candidate Gene ID | Description |
|---|---|---|---|---|---|
| *qfsd_Ah11* | Ah11 | Ah11_8690364 | Ah11_17741529 | Arahy.62XZQ4 | WRKY family transcription factor family protein |
| | | | | Arahy.SS16UG | Zinc finger CCCH-type with G patch domain protein |
| | | | | Arahy.CW9Q3P | Histone deacetylase 1 |
| | | | | Arahy.XCJ8AH | Receptor-like protein kinase 2-like (Leucine-rich repeat) |
| | | | | Arahy.G0S5C3 | Eukaryotic aspartyl protease family protein |
| | | | | Arahy.XA421G | Serine/threonine-protein kinase SRK2I-like isoform 1 |
| | | | | Arahy.40JS6F | C-terminal domain phosphatase-like 4 |
| | | | | Arahy.DB7AVX | Cytochrome P450, family 711, polypeptide 1 |
| *qfsd_Ah06* | Ah06 | Ah06_112890279 | Ah06_109915576 | Arahy.U8SEXI | Histone deacetylase 2 |
| | | | | Arahy.HTBE6N | Receptor-like protein kinase 2 |
| | | | | Arahy.PS92UU | Protein phosphatase 2C family protein |
| | | | | Arahy.38FA9N | Protein phosphatase 2C family protein |
| | | | | Arahy.6J2YJR | MYB transcription factor MYB60 |
| *qfsd_Ah16* | Ah16 | Ah16_141375358 | Ah16_138472734 | Arahy.E15ZAA | Receptor-like kinase 1 (Leucine-rich repeat) |
| | | | | Arahy.AP90LP | Histone-lysine N-methyltransferase ATX5-like |
| | | | | Arahy.VVF851 | Protein kinase superfamily protein |
| | | | | Arahy.J7EN8P | E3 ubiquitin-protein ligase BRE1-like protein |
| *qfsd_Ah17* | Ah17 | Ah17_69164615 | Ah17_2853004 | Arahy.6JQ8YQ | Cytochrome P450 superfamily protein |
| | | | | Arahy.7BG7FU | Cytochrome P450 superfamily protein |
| | | | | Arahy.EH049N | Cytochrome P450 superfamily protein |
| | | | | Arahy.ETWN8F | Cytochrome P450 superfamily protein |
| | | | | Arahy.9E5BV9 | Cytochrome P450 superfamily protein |
| | | | | Arahy.JE5MND | Cytochrome P450 superfamily protein |
| | | | | Arahy.H7RCEW | C2H2-like zinc finger protein |
| | | | | Arahy.3516XA | Serine/threonine protein phosphatase 2A |

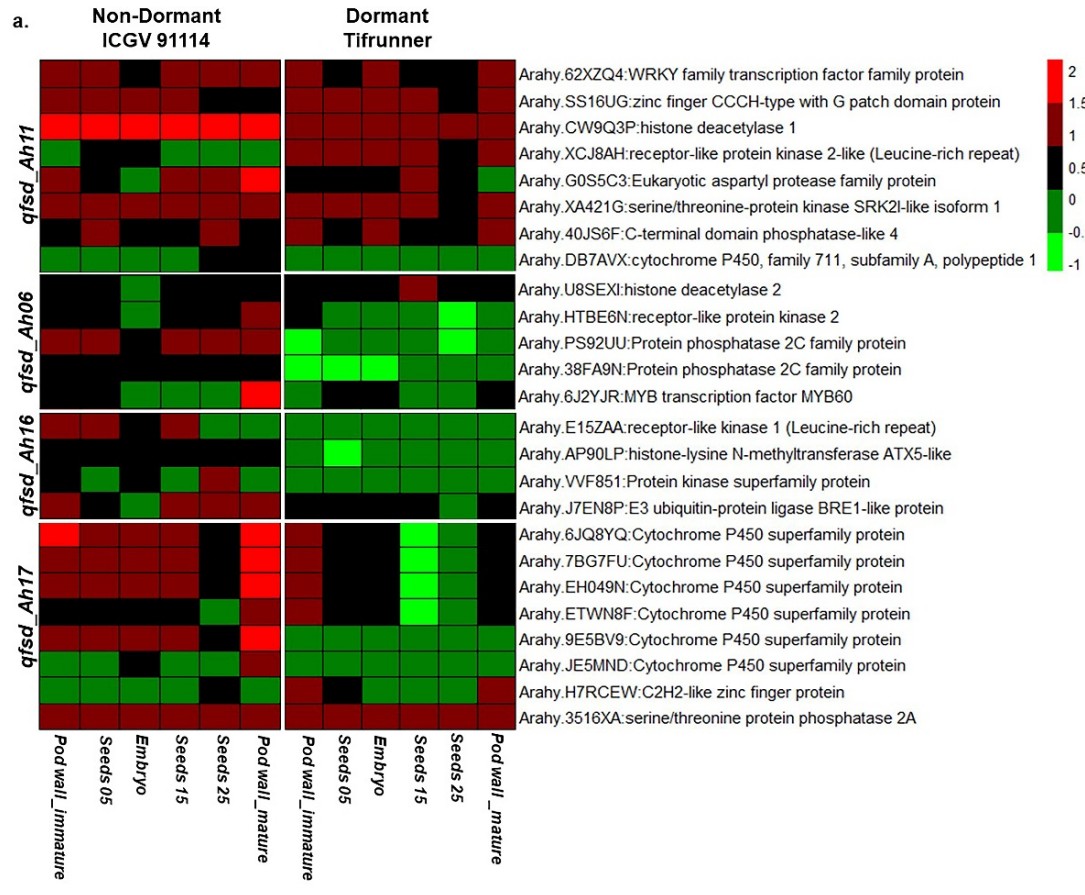

**Figure 4.** *Cont*.

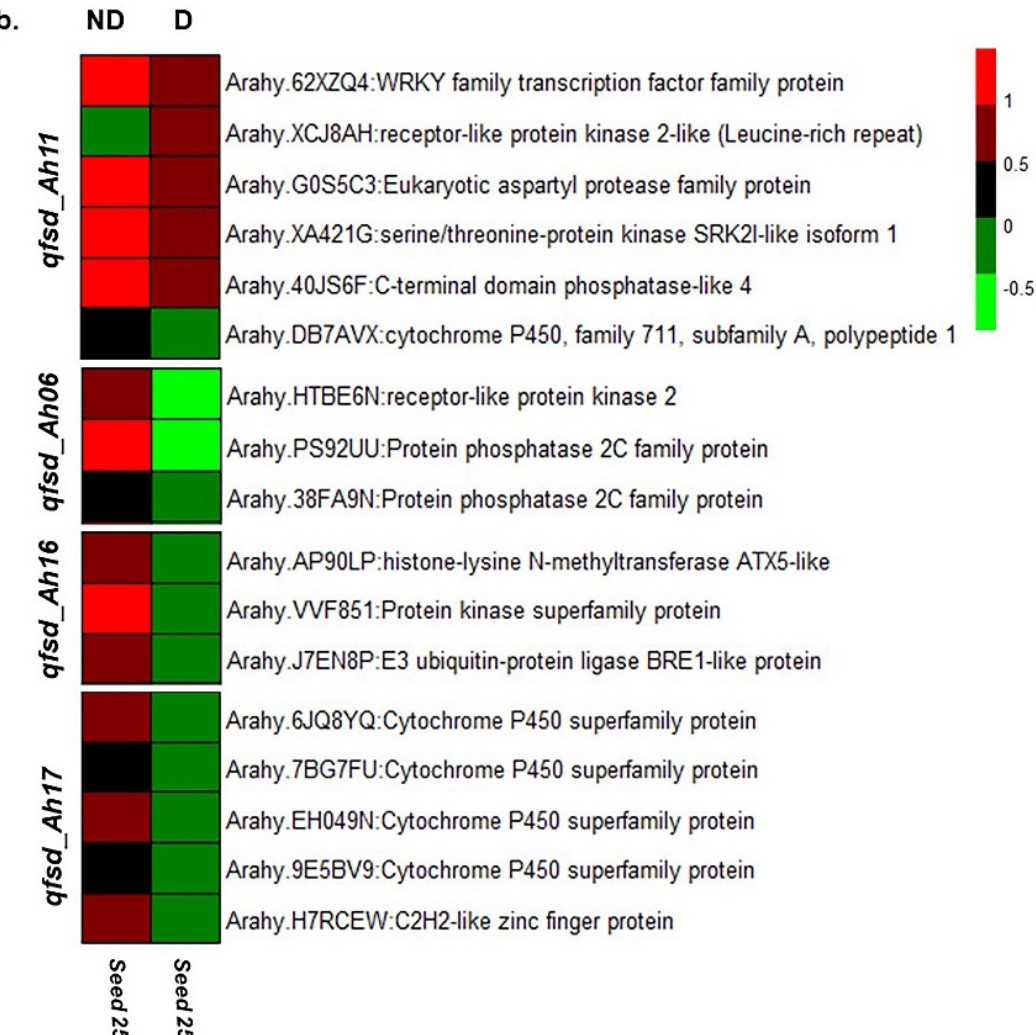

**Figure 4.** Heat map showing differentially expressed genes between dormant Tifrunner and non-dormant ICGV 91114 genotypes. The expression of selected genes in identified major QTLs during (**a**) seed and pod development stages and (**b**) only at seed_25 stage (mature seed) between dormant and non-dormant genotype.

Based on a previous sequencing-based trait mapping approach, 13 SNPs which were located in the genic regions of candidate genes were selected for development of KASP markers (Table S6). From 13 KASPs, 8 KASPs were selected from chromosome A09, 2 KASPs from A10 and 3 KASPs from B05. These 13 KASP markers were validated for polymorphism in dormant and non-dormant germplasm lines and selected RILs of the ICGV 02266 × ICGV 97045 population. Of these 13 KASPs, a total of 6 KASP markers were validated successfully with clear discrimination between dormant and non-dormant groundnut genotypes (Figure 5).

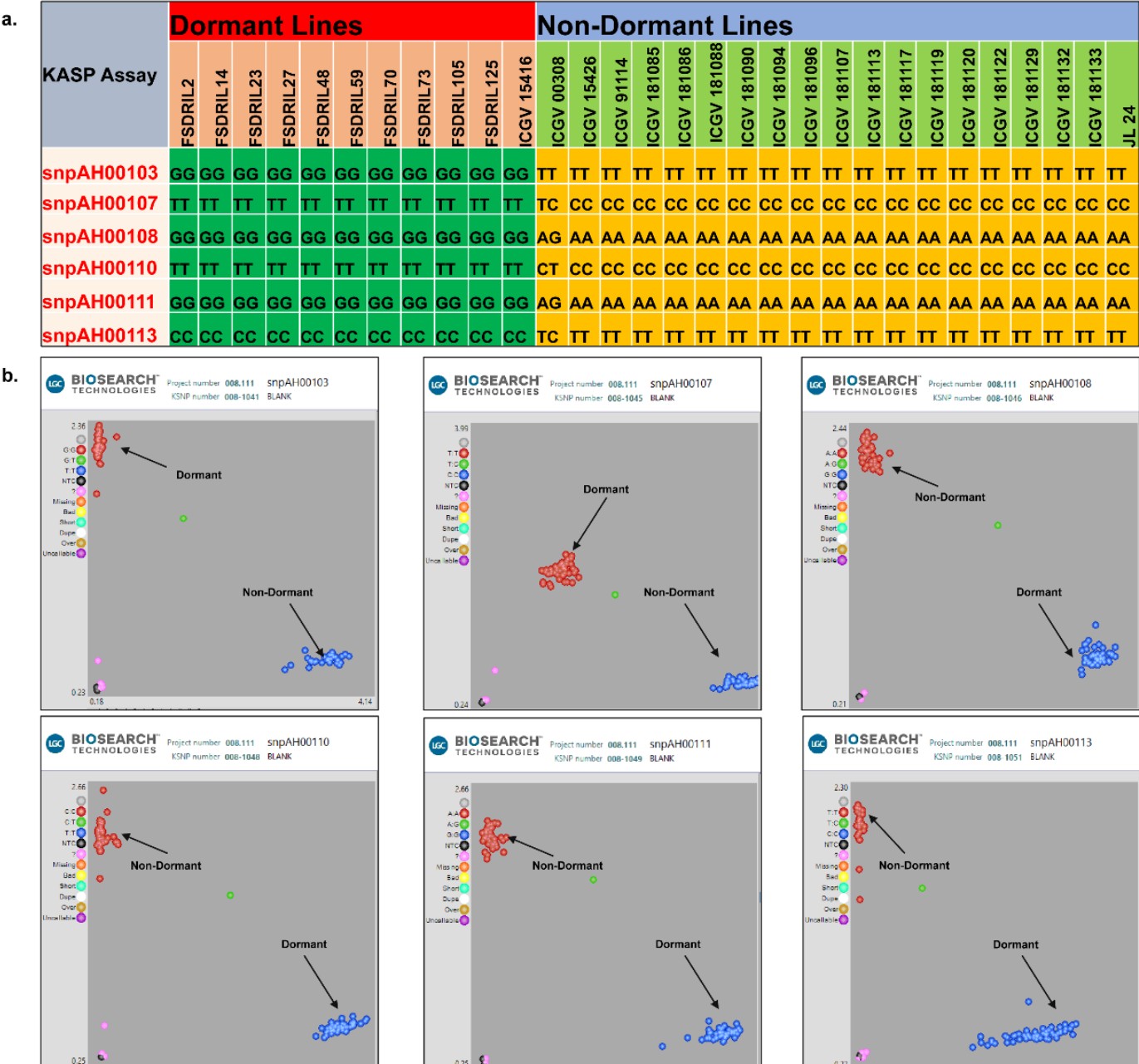

**Figure 5.** Validation of KASP assays on dormant and non-dormant groundnut genotypes. (**a**) SNP calls and (**b**) Cluster plot distribution across a diverse panel of dormant and non-dormant genotypes.

## 4. Discussion

Groundnut is the major oilseed crop, mainly cultivated in tropical, subtropical and temperate regions of the world under rainfed conditions. Spanish cultivars are the most widely grown types of groundnut but lack dormancy, unlike the Virginia genotypes. Untimely rains and prevailing wet conditions during crop maturity lead to pre-harvest sprouting of seeds, hampering yield and seed quality. So, the presence of FSD for 2–3 weeks in commercial Spanish cultivars helps reduce losses due to pre-harvest sprouting and contributes to better adoption and sustainable groundnut production. GAB offers a comparative advantage over conventional breeding by efficient tracing of alleles among segregating lines using diagnostic markers. Therefore, our study has been conducted to identify the genomic regions and candidate genes regulating FSD, which further assists in developing diagnostic markers.

Highly informative fixed SNP arrays can be profitably utilized for diversity analysis, QTL mapping and association studies as they can genotype a large number of samples in a short time period [32]. Several high and mid-density SNP assays have been developed and successfully deployed in wheat [33], rice [34,35], maize [36], groundnut [13,15] and soybean [37]. Using a mid-density 5 K SNP assay, we genotyped a RIL population of ICGV 02266 × ICGV 97045. Our study reports the development of a high-quality consensus linkage map of 325 SNP loci spanning 2335.3 cM map distance with an average inter-marker distance of 7.2 cM per locus. Earlier, an 'Axiom_*Arachis*' 58 K SNP array was used for genotyping a RIL population and constructing a high-density genetic map with 1147 SNP loci spanning 1679.72 cM map length and 1.58 cM per loci map density [16].

In the present study, a total of five major FSD QTLs were identified on Ah01, Ah11, Ah06, Ah16 and Ah17 chromosomes using ICIM and QTL Cartographer software's with a relatively higher LOD and PVE detected in ICIM than in QTL cartographer. Major QTLs were identified on homeologous Ah01/Ah11 and Ah06/Ah16 chromosomes. Previously, two dormancy QTLs on A05 and B06 chromosomes were reported in a $F_2$ population (ICGV 00350 × ICGV 97045) [3], but later two QTLs were reported on B05 and A09 using QTL-seq in an RIL population (ICGV 00350 × ICGV 97045) [12]. These results also indicated that the genomic regions associated with seed dormancy were identified on homeologous chromosomes A05 and B05. Since the reference genome for cultivated tetraploid was released in 2019, these previous studies did not have access to a cultivated genome for precise genome location and candidate genes. Nevertheless, the genome sequence of allotetraploid groundnut (*Arachis hypogaea* cv. Tifrunner) was reported to show correspondence between the homeologous chromosome pairs [17]. During the course of genome evolution, recombination between corresponding homeologous chromosomes led to genetic exchange between non-sister chromatids of homeologous chromosomes [38]. Therefore, because of genome similarity between the homeologous chromosomes of the two progenitor genomes (A and B genome) of *A. hypogaea*, the homeologous associations/QTLs are identified. Similar results were reported in other allopolyploids like wheat, where QTLs for seed dormancy were identified on homeologous chromosomes 3A/3B [39–41], 4A/4B and 5A/5B [42], 1A and 1B, 7A and 7B [43]. Our study also identified a large number (a total of 103) of epistatic QTLs with higher LOD and PVE% on homeologous chromosomes. Earlier studies on the inheritance of FSD in groundnut reported that the trait is governed by a single dominant gene [24] and two duplicative recessive genes [3,4]. Furthermore, additive and dominance effects reported duplicate epistasis controlling FSD by generating three separate crosses from Spanish varieties [44].

The role of ABA signaling and its interactions with GAs and ethylene in modulating the seed transition from dormancy to germination are critical [45]. Genes reported to be involved in the ABA/GA/ethylene biosynthesis pathways identified from the genomic regions of major QTLs were carried forward for differential gene expression analysis between dormant (Tifrunner) and non-dormant (ICGV 91114) groundnut genotypes. As a result, 25 genes that showed contrasting expression between dormant and non-dormant groundnut genotypes were identified as the potential candidates having functional relevance for dormancy. Of these 25 potential candidate genes, multiple copies of *cytochrome P450 super family protein* genes were identified in the '*qfsd_Ah11*' and '*qfsd_Ah17*' genomic regions. Other promising genes like *histone deacetylase 1* (*Arahy.CW9Q3P*) on '*qfsd_Ah11*' and *histone deacetylase 2* (*Arahy.U8SEXI*) on '*qfsd_Ah06*' appeared as the key regulators of ABA signaling [46,47]. *Ethylene responsive transcription factors* were identified on QTL regions of Ah01, Ah06 and Ah17 chromosomes, but their expression levels were reported only in ICGV 91114. Ethylene biosynthesis antagonizes ABA signaling pathways and promotes seed germination in many crop species [48]. Several *receptor-like kinases* have been identified on Ah11, Ah06 and Ah16 chromosomes which are involved in activating ABA responsive genes regulating seed germination under abiotic stress conditions [49].

In the QTL region of '*qfsd_Ah11*', we noticed *histone deacetylase 1* (*Arahy.CW9Q3P*) exhibited comparatively higher expression levels in all the selected tissues of non-dormant

ICGV 91114, indicating its negative regulatory role in dormancy. Histone deacetylation has been linked to the suppression of positively regulating seed germination genes [47]. In Arabidopsis, *Histone deacetylase 19* (*HDA19*) interacts with *SIN3-LIKE1* (*SNL1*) and represses seed germination inducing genes like *ethylene response factors* through histone deacetylation to maintain normal dormancy levels [50]. On the contrary, dormant accessions of Arabidopsis exhibited a lower level of *Histone Deacetylase 2B* (*HD2B*), depicting its negative role in dormancy [46]. *Aspartic protease ASPG1* (*Aspartic Protease in Guard Cell 1*) of Arabidopsis was reported to be associated with degradation of seed storage proteins (SSP) required for the seed germination process. *aspg1−1* mutants showed enhanced dormancy by increasing the expression of DELLA proteins, which are known to suppress GA signal transduction pathways [51]. In accordance with this, our study also noticed higher expression of *eukaryotic aspartyl protease* family protein (*Arahy.G0JS6F*) in seeds_25 and the pod wall of non-dormant ICGV 91114, supporting the role of the seeds and pod wall in regulating germination in groundnut. The wild-type *ABI1* gene of Arabidopsis encodes *serine/threonine phosphatases* (*PP2C*), a negative regulator of ABA signaling [52]. In the present study, *serine/threonine protein phosphatase SRK2I-like isoform 1* (*Arahy.XA421G*) showed a comparatively high expression in ICGV 91114, indicating its negative role in ABA signaling. *Zinc finger CCCH-type with G patch domain protein* (*Arahy.SS16UG*), a prominent gene on chromosome Ah11, showed a relatively high expression pattern in the mature pod wall of the dormant Tifrunner. Similarly, in Arabidopsis, a mutant of *CCCH-type zinc finger protein* showed lower ABA and higher GA levels by regulation of GA and ABA signaling pathways via *Phytochrome-Interacting Factor3-Like5* (*PIL5*), indicating its positive role in dormancy [53]. As FSD is a complex trait, several transcription factors or activators were also identified in this study as playing an important role in the regulation of gene expression. *WRKY transcription family protein* (*Arahy.62XZQ4*) showed comparatively higher expression in seeds of non-dormant ICGV 91114 than in dormant Tifrunner in mature and imbibed seeds of Arabidopsis, lack of *WRKY transcription factor 41* (*WRKY41*) also showed decreased *Abscisic acid insensitive 3* (*ABI3*) [54].

In the QTL region of 'qfsd_Ah06', we noticed slightly higher expression of *histone deacetylase 2* (*Arahy.U8SEXI*) in the embryo and seeds_15 of the dormant Tifrunner genotype, indicating its positive role in dormancy. Compared with previous reports, it is clear that *histone deacetylase* (*HDAC*) affects dormancy either negatively (*HD2B*) or positively (*HDA19*), depending on the type of gene they act on and their mechanisms. In Arabidopsis, *MYB96 transcription factor* positively regulates seed dormancy by promoting ABA biosynthesis *NCED* genes and down regulating *GA3ox1* and *GA20ox1* (GA biosynthesis genes) [55]. Mutant seeds of *myb96−1* showed earlier germination than the wild-type (*MYB96−1*). In the present investigation, higher expression of *MYB60* (*Arahy.6J2YJR*) was observed in the embryo of the dormant Tifrunner, whereas in non-dormant ICGV 91114, the expression was higher in the pod wall. *RDO5* (*Reduced dormancy 5*) encodes *protein phosphate phosphatase 2C*, reported to be involved in ABA signaling. A mutant *rdo5* with reduced seed dormancy isolated from mutagenesis screening showed enhanced transcript levels of *APUM9* and *APUM11* [56]. The present study also noticed higher levels of *protein phosphatase 2C family proteins* in non-dormant ICGV 91114, denoting its positive role in germination.

On 'qfsd_Ah16', *histone-lysine N-methyltransferase ATX5-like* (*Arahy.AP80LP*) showed high expression during different seed and pod development tissues of ICGV 91114, making it a positive regulator of germination. In imbibed seeds of Arabidopsis, *SUVH5* and *SUVH4* (*a histone H3 lysine 9 methyltransferase*) suppressed ABA biosynthesis genes, revealing its positive role in seed germination [57,58]. Protein ubiquitination by E3 ligases plays a major role in several plant developmental stages including dormancy and germination [59]. In the present study, E3 ubiquitin-protein ligase *BRE1-like protein* (*Arahy.J7EN8P*) has shown contrasting gene expression in dormant (Tifrunner) and non-dormant (ICGV 91114) where the seeds and pod wall of non-dormant ICGV 91114 have shown higher expression levels.

On 'qfsd_Ah17', multiple copies of *cytochrome P450 superfamily protein genes* showed high expression values in non-dormant ICGV 91114, depicting its negative role in regulating

dormancy. Similarly, another copy of the *cytochrome P450 protein* (*Arahy.DB7AVX*) on '*qfsd_Ah11*' also showed a higher expression pattern in the seed and pod wall of ICGV 91114. In Arabidopsis, *CYP707A*, a *cytochrome P450 superfamily protein* is reported to be involved in ABA catabolism by ABA *8′-hydroxylation*. Expression analysis revealed that mutant *cyp707a2* showed six-fold higher ABA, exhibiting hyper seed dormancy than the wild types [60]. *Serine/threonine protein phosphatase 2A* (*Arahy.3516XA*) on '*qfsd_Ah17*' also showed comparatively high expression in non-dormant ICGV 91114 indicating its negative role in ABA signaling.

In the present study, differential expression levels of receptor like protein kinases were observed on the Ah06, Ah11, and Ah16 chromosomes. *Receptor-like protein kinase 2-like* (*Leucine-rich repeat*) on Ah11 showed high expression in the dormant Tifrunner. But the receptor like *protein kinase 1* and *2* on Ah06 and Ah16 chromosomes respectively, showed high expression in non-dormant ICGV 91114. Reduced seed germination was reported in overexpressing *FON1* (*a leucine-rich receptor-like kinase*) lines of rice [61]. Similarly, several protein kinases like *SnRK1A*, *OSRK1* are reported to be involved in the regulation of seed germination and seedling establishment in rice [49,50].

By considering this comparative gene expression analysis, *histone deacetylases, cytochrome P450 superfamily protein, ethylene-responsive transcription factor* and *histone-lysine N-methyltransferase ATX5-like* seem to be the potential regulators involved in the pathways associated with dormancy or germination. From our findings, we presume that dormancy and germination are two different traits where repression of one trait and suppression of the other could lead to either dormancy or germination.

The use of a KASP marker helps in effective selection of desirable individuals for desired traits [62]. Our investigation showed that the association of favourable alleles was reliable for six KASP markers, suggesting the markers' usefulness in screening germplasm and will help in pyramiding of linked favourable alleles for FSD in groundnut.

## 5. Conclusions

This study deployed cost-effective 5 K SNP genotyping assay for genetic map construction and QTL identification. Five major QTLs with higher LOD and PVE % and several genes with associated function have been identified. Furthermore, six KASP markers were validated in a diverse panel for conclusive allele information which will be further utilized in genomic assisted breeding to improve the fresh seed dormancy trait in groundnut.

**Supplementary Materials:** The following supporting information can be downloaded at: https://www.mdpi.com/article/10.3390/agronomy12020404/s1, Table S1: Validation panel for validating the KASP markers for fresh seed dormancy in groundnut. Table S2: List of major QTLs identified for FSD in ICGV 02266 × ICGV 97045 using WinQTL Cartographer version 2.5 and IciMapping version 4.2. Table S3: List of epistatic QTLs (Q × Q) identified for FSD in ICGV 02266 × ICGV 97045 using IciMapping version 4.2. Table S4: List of candidate genes having functional relevance with dormancy and germination identified from genomic regions of the five major QTLs. Table S5: Expression values of the genes showing contrasting expression in the seed and pod tissues of ICGV 91114 (non-dormant) and Tifrunner (dormant). Table S6: Details of KASP assays developed and validated for fresh seed dormancy.

**Author Contributions:** M.K.P. conceived the idea, supervised the study and finalized the manuscript; P.J., M.T.V., S.S.G., D.B., S.S.M. and D.D. developed and phenotyped RIL population; D.B. performed genotyping, genetic mapping, QTL analysis, candidate gene discovery and drafting of manuscript; S.S.G. and V.S. contributed to DNA isolation, genotyping, analysis related to gene expression, improving figures and manuscript; S.P. contributed to QTL analysis; R.K.V., B.G., M.R., P.S., D.M.R. and B.V.B.R. contributed to interpretations and improving manuscript. All authors have read and agreed to the published version of the manuscript.

**Funding:** This research was funded by the National Agricultural Science Fund (NASF) of the Indian Council of Agricultural Research (ICAR), India.

**Institutional Review Board Statement:** Not Applicable.

**Informed Consent Statement:** Not Applicable.

**Data Availability Statement:** The data and detailed results are provided in supplementary files.

**Acknowledgments:** The authors thank the National Agricultural Science Fund (NASF) of the Indian Council of Agricultural Research, India for partial financial assistance in this study. Deekshitha Bomireddy acknowledges Acharya N.G. Ranga Agricultural University for collaborating with ICRISAT and the opportunity given as a student to pursue this investigation at ICRISAT. The work was carried out as part of CRP-GLDC. The authors are thankful for the support from OFID for the population development and phenotyping work.

**Conflicts of Interest:** The authors declare there is no conflict of interest.

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
