# Peer review of "Discovery of Major Quantitative Trait Loci and Candidate Genes for Fresh Seed Dormancy in Groundnut"

_agronomy, doi:10.3390/agronomy12020404_

Round 1

Reviewer 1 Report

The authors developed the RIL population and genotyped them with 5K SNPs assay. Further, authors discovered major quantitative trait loci about fresh seed dormancy in groundnut based on this linkage map. Finally, several potential candidate genes were identified from the major QTL regions whose differential gene expression was established using transcriptomic information.The content of this experiment is detailed and logical, and has certain practicability. However, some modifications are still needed.

The specific recommendations are as follows:

  1. The results of this study show that quantitative trait locus analysis identified five major main-effect QTLs, but only one (qfsd_Ah17) was consistently detected in two seasons. I recommended authors to focus on this QTL region (qfsd_Ah17) and identified potential candidate genes about fresh seed dormancyfrom more transcriptomic information. Therefore, I recommended authors to add more transcriptomic information to screen the candidate genes about fresh seed dormancy from this QTL region (qfsd_Ah17).

Author Response

Response to the comments of the Reviewer#1

The authors developed the RIL population and genotyped them with 5K SNPs assay. Further, authors discovered major quantitative trait loci about fresh seed dormancy in groundnut based on this linkage map. Finally, several potential candidate genes were identified from the major QTL regions whose differential gene expression was established using transcriptomic information. The content of this experiment is detailed and logical, and has certain practicability. However, some modifications are still needed.

The specific recommendations are as follows:

  1. The results of this study show that quantitative trait locus analysis identified five major main-effect QTLs, but only one (qfsd_Ah17) was consistently detected in two seasons. I recommended authors to focus on this QTL region (qfsd_Ah17) and identified potential candidate genes about fresh seed dormancy from more transcriptomic information. Therefore, I recommended authors to add more transcriptomic information to screen the candidate genes about fresh seed dormancy from this QTL region (qfsd_Ah17).

Authors’ Response:  We thank the Reviewer for this suggestion. We have updated the transcriptomic information on serine-threonine phosphatase which is considered as an important candidate gene governing dormancy. As this QTL region (qfsd_Ah17) is consistent in all the three seasons, we are planning for further fine mapping and identification of causal genes/nucleotides associated with dormancy.

Reviewer 2 Report

Dear authors,

I found your paper clearly written with lots of data well presented in figures and tables. The introduction has enough information about the scientific discoveries regarding your subject. The Material and Methods you have used in the research are well described and the results are clearly presented. Also, the figures and tables are enriching the quality of the manuscript. 

I only have few recommendations:

The last paragraph from the Discussion section is more like a conclusion (lines 587-597).

If you can improve some of the cited articles, it will improve the percentage of current articles cited in your manuscript. At this moment, 60% of them are older than 5 years. 

The first paragraph of Conclusion (600-607)  looks like an abstract of the experiment.  

Author Response

Response to the comments of the Reviewer#2

I found your paper clearly written with lots of data well presented in figures and tables. The introduction has enough information about the scientific discoveries regarding your subject. The Material and Methods you have used in the research are well described and the results are clearly presented. Also, the figures and tables are enriching the quality of the manuscript.

Authors’ Response: Thank you for kind appreciation of our research work. We are grateful to you for providing valuable suggestion and important comments.

I only have few recommendations:

The last paragraph from the Discussion section is more like a conclusion (lines 587-597).

Authors’ Response: Thanks for your suggestion. We have now corrected these lines in the text.

If you can improve some of the cited articles, it will improve the percentage of current articles cited in your manuscript. At this moment, 60% of them are older than 5 years.

Authors’ Response: Thanks for your suggestion. Please note that articles on aspect of FSD is limited in groundnut, we have tried to provide all current articles citation in the manuscript.

The first paragraph of Conclusion (600-607) looks like an abstract of the experiment. 

Authors’ Response: Thanks for this suggestion and we have now updated the Conclusion section and provided information on future implication of this study.

Reviewer 3 Report

Dear Editor, Dear Authors,

I am glad to be considered as a reviewer for the manuscript entitled: "Discovery of Major Quantitative Trait Loci and Candidate Genes Controlling Fresh Seed Dormancy in Groundnut". The manuscript presents the results of very interesting research aimed at identifying genomic regions related to the dormancy state of fresh peanut seeds (Arachis hypogaea L.). The clear tendency towards pre-sowing germination of seeds of this species actually causes a significant reduction in the yield as well as the final quality of the seeds. Currently, biotechnological tools based on molecular studies are a very good way to improve and accelerate the cultivation of crops. The results presented in this manuscript can undoubtedly be used in peanut breeding practice, as they will facilitate the selection of genotypes resistant to in-situ germination or pre-harvest sprouting of seeds and accelerate the creation of new and improved cultivars of this species.

However, I have a few small remarks that I hope will help to improve this manuscript:

1) Please consider shortening the Abstract.

2) The Keywords should not repeat the words from the title of the manuscript, and the keywords should be arranged in alphabetical order.

3) The manuscript is very poorly suited to the requirements of the Agronomy journal template; particular attention should be paid to Table formatting and the References chapter.

4) Tables 2 and 3 are poorly readable and incomprehensible; all abbreviations used in the Table should be explained either under the table or in the Table title.

Author Response

Response to the comments of the Reviewer#3

Dear Editor, Dear Authors,

I am glad to be considered as a reviewer for the manuscript entitled: "Discovery of Major Quantitative Trait Loci and Candidate Genes Controlling Fresh Seed Dormancy in Groundnut". The manuscript presents the results of very interesting research aimed at identifying genomic regions related to the dormancy state of fresh peanut seeds (Arachis hypogaea L.). The clear tendency towards pre-sowing germination of seeds of this species actually causes a significant reduction in the yield as well as the final quality of the seeds. Currently, biotechnological tools based on molecular studies are a very good way to improve and accelerate the cultivation of crops. The results presented in this manuscript can undoubtedly be used in peanut breeding practice, as they will facilitate the selection of genotypes resistant to in-situ germination or pre-harvest sprouting of seeds and accelerate the creation of new and improved cultivars of this species.

Authors’ Response: We thank the Reviewer for kind appreciation of our research work. We are grateful to him/her for providing valuable suggestions and important comments.

However, I have a few small remarks that I hope will help to improve this manuscript:

1) Please consider shortening the Abstract.

Authors’ Response: Thanks for this important suggestion. We have reduced text in the abstract and kept minimum important description for the benefit and clarity to the readers in this section

2) The Keywords should not repeat the words from the title of the manuscript, and the keywords should be arranged in alphabetical order.

Authors’ Response: Thank you for this important suggestion. We have updated the keywords as per your suggestions.

3) The manuscript is very poorly suited to the requirements of the Agronomy journal template; particular attention should be paid to Table formatting and the References chapter.

Authors’ Response: Thanks for bringing this to our attention. We have thoroughly gone through the manuscript and corrected the typo/format mistakes.

4) Tables 2 and 3 are poorly readable and incomprehensible; all abbreviations used in the Table should be explained either under the table or in the Table title.

Authors’ Response: Thank you for this important suggestion. We have updated the tables and provided the explanation for all abbreviations to avoid confusion.

In addition, we have included 3rd season QTL results as well as validation of KASP markers from our previous study in the revised version of MS

We hope that the MS has significantly improved based on the suggestions and comments of three Reviewers, and is ready for acceptance and publication